# Bioconversion Process of Polyethylene from Waste Tetra Pak^®^ Packaging to Polyhydroxyalkanoates

**DOI:** 10.3390/polym14142840

**Published:** 2022-07-12

**Authors:** Itohowo Ekere, Brian Johnston, Fideline Tchuenbou-Magaia, David Townrow, Szymon Wojciechowski, Adam Marek, Jan Zawadiak, Khadar Duale, Magdalena Zieba, Wanda Sikorska, Grazyna Adamus, Tomasz Goslar, Marek Kowalczuk, Iza Radecka

**Affiliations:** 1School of Science, Faculty of Science and Engineering, University of Wolverhampton, Wolverhampton WV1 1LY, UK; i.a.jonah@wlv.ac.uk (I.E.); d.townrow@wlv.ac.uk (D.T.); 2Science in Industry Research Centre (SIRC), SciTech Innovation Hub, Wolverhampton Science Park, Glaisher Drive, Wolverhampton WV10 9RU, UK; b.johnston2@wlv.ac.uk; 3School of Engineering, Computing and Mathematical Sciences, Faculty of Science and Engineering, University of Wolverhampton, Wolverhampton WV1 1LY, UK; f.tchuenbou-magaia@wlv.ac.uk; 4Department of Chemical Organic Technology and Petrochemistry, Silesian University of Technology, 44-100 Gliwice, Poland; szymonwoj@gmail.com (S.W.); adam.a.marek@polsl.pl (A.M.); jan.zawadiak@polsl.pl (J.Z.); 5Centre of Polymer and Carbon Materials Polish Academy of Sciences, 34 M. Curie-Sklodowska St., 41-800 Zabrze, Poland; kduale@cmpw-pan.edu.pl (K.D.); mzieba@cmpw-pan.edu.pl (M.Z.); wanda.sikorska@cmpw-pan.edu.pl (W.S.); grazyna.adamus@cmpw-pan.edu.pl (G.A.); 6Poznan Radiocarbon Laboratory, Poznan Park of Science and Technology, 46 Rubiez St., 61-612 Poznan, Poland; goslar@radiocarbon.pl

**Keywords:** Tetra Pak^®^, *Cupriavidus necator*, polyhydroxyalkanoate (PHA), bioplastics, recycling, polyethylene, bioconversion, sustainability, C14 analysis

## Abstract

Presented herein are the results of a novel recycling method for waste Tetra Pak^®^ packaging materials. The polyethylene (PE-T) component of this packaging material, obtained via a separation process using a “solvents method”, was used as a carbon source for the biosynthesis of polyhydroxyalkanoates (PHAs) by the bacterial strain *Cupriavidus necator* H16. Bacteria were grown for 48–72 h, at 30 °C, in TSB (nitrogen-rich) or BSM (nitrogen-limited) media supplemented with PE-T. Growth was monitored by viable counting. It was demonstrated that *C. necator* utilised PE-T in both growth media, but was only able to accumulate 40% *w/w* PHA in TSB supplemented with PE-T. Only 1.5% *w/w* PHA was accumulated in the TSB control, and no PHA was detected in the BSM control. Extracted biopolymers were characterised by nuclear magnetic resonance (NMR), Fourier-transform infrared (FTIR) spectroscopy, electrospray tandem mass spectrometry (ESI-MS/MS), gel permeation chromatography (GPC), and accelerator mass spectrometry (AMS). The characterisation of PHA by ESI-MS/MS revealed that PHA produced by *C. necator* in TSB supplemented with PE-T contained 3-hydroxybutyrate, 3-hydroxyvalerate, and 3-hydroxyhexanoate co-monomeric units. AMS analysis also confirmed the presence of 96.73% modern carbon and 3.27% old carbon in PHA derived from Tetra Pak^®^. Thus, this study demonstrates the feasibility of our proposed recycling method for waste Tetra Pak^®^ packaging materials, alongside its potential for producing value-added PHA, and the ability of 14C analysis in validating this bioconversion process.

## 1. Introduction

Tetra Pak^®^ is a contemporary type of packaging widely used around the world for aseptically packaged goods. Billions of litres of liquid food items, such as milk and juice, require Tetra Pak^®^ to maintain their freshness and maximise their quality without any preservatives [1]. Tetra Pak^®^ packaging materials are made up of three primary raw materials: paperboard, aluminium foil, and low-density polyethylene (LDPE) [1]. The paperboard is the dominant material in Tetra Pak^®^ cartons; it constitutes 75% of the total carton weight, where it provides mechanical strength and stability to the packaging [1,2]. The LDPE functions as an adhesive agent between the components of the packaging material, and also protects the product from external moisture, while making up 20% of the total packaging weight [1]. Despite comprising only 5% of the packaging, the aluminium plays a vital role in the thermal stabilisation of the food product by blocking out oxygen and light [2]. Therefore, it inhibits both microbial growth (which could promote oxidation damage) and photodegradation, which causes the loss of vital vitamins (A, B_2_, C, D, and E), carboxylic acids, enzymes, and the food’s flavour [2]. Blocking UV light is also particularly important for dairy products, as riboflavin is susceptible to UV light degradation [3,4,5].

Tetra Pak^®^ packages are used in more than 160 countries around the world [6]. In 2019, over 190 billion Tetra Pak^®^ packages were sold [6]. The geography of net sales indicated that 89% of total sales happened in Europe, America, and Asia [6]. In light of these statistics, it is certain that the amount of waste Tetra Pak^®^ in municipal solid waste is also increasing considerably, especially given the fact that the majority of these packages are dumped in landfills after use [7]. The year 2016 recorded the accumulation of 188 billion tonnes of waste Tetra Pak^®^, which is expected to continually increase annually [8]. Of this waste Tetra Pak^®^, only a small amount is recycled, while the rest is incinerated to toxic ash or piled up on beaches, roads, rubbish dumps, and landfills [9]. For instance, the global recycling rate of Tetra Pak^®^ packages was only 26% in 2019 [10]. It is known that this accumulation in landfills and incineration can lead to environmental pollution, but more importantly, the polyethylene polymer within the packaging material is believed to cause negative toxic effects on the health of humans and animals, affecting the endocrine system, and linked to cancer [11] and reproductive problems [12].

Current recycling techniques can be broadly separated into three main categories: One category involves methods that do not separate out the three components, and either has energy recovery as the primary target, or converts everything into other materials, such as chipboard products, fillers, roof sheets, and polymer concrete [13,14,15]. Another category uses the hydro-pulping technique, which involves the mechanical separation of paperboard from LDPE and aluminium, leaving behind a combined LDPE–aluminium component [14,15]. This seems to be the main recycling route, and a large amount of cellulose fibre is recovered, which can then be reused for the manufacturing of paper and packaging materials [14]. The recycled LDPE and aluminium (so-called recycled polyAl) can either be converted into other useful products—such as panel boards, roof sheets, waterproof boards, and furniture—or can be used for energy recovery through incineration [14,15]. Methods of the third category, which recover aluminium through a thermochemical process such as pyrolysis [16,17], or through the separation of polyethylene and aluminium by acid-based wet processing techniques and plasma technology, have also been reported [18,19].

Despite all of these recycling solutions, there still exist some major challenges that limit the full recycling of Tetra Pak^®^. Among these challenges are the lack of adequate Tetra Pak^®^ recycling infrastructure in most parts of the world, the high processing cost of the recycling methods, high energy consumption, low demand for recycled end products, and low engagement and the lack of education on post-consumer recycling [7]. Considering the abovementioned recycling problems, research is still ongoing to improve the life cycle of Tetra Pak^®^ packaging by enhancing the recycling process of waste Tetra Pak^®^ packages—in particular, by upcycling these wastes to valuable products with economic and environmental importance [13,19,20,21]. One such effort is to find a way to recover the LDPE and return it into the carbon cycle. This paper reports a novel recycling technique of LDPE from Tetra Pak^®^ waste to synthesise polyhydroxyalkanoates (PHAs) via *Cupriavidus necator.* PHAs are biopolyesters that can be synthesised from a wide range of carbon sources by many microbial strains. They are viewed as a class of biodegradable bioplastics that can potentially replace synthetic petroleum-derived plastics and minimise the hazardous impacts caused by synthetic plastic waste. PHA production from waste Tetra Pak^®^ could be an important economical alternative to the other recycling methods of polyethylene polymer materials. We have extensively described the use of oxidised polypropylene (PP) as a feedstock for PHA production in previous publications [22], as well as a recycling method for controlled oxidative fragmentation of LDPE plastics by *Cupriavidus necator* [23]. The PHB producer *Cupriavidus necator* consistently stands out for its ability to produce PHA from multiple carbon sources, including sugars and fatty acids [23]. It is genetically stable, and is one of the most studied microbes used in this area of study, making it a potential candidate for industrial PHA production [22,23]. This strain was used as a biocatalyst to upcycle LDPE fragments in Tetra Pak^®^ waste to value-added PHA. The present study reveals the effects of the waste Tetra Pak^®^ mechanical grinding and separation process, using the “solvents method” of PE, as a possible carbon source for the biosynthesis of high-value biodegradable PHA.

## 2. Materials and Methods

### 2.1. Carbon Source

Polyethylene from Tetra Pak^®^ packaging was purified according to the Polish patent in [24]. A schematic diagram of the method used for the processing of polyAl from waste Tetra Pak^®^ is shown in Figure 1. Waste Tetra Pak^®^ packages were processed in an industrial paper mill suited for the handling of such materials. The process resulted in above 95% paper fibre recovery. PolyAl, obtained as a byproduct, was subjected to magnetic and pneumatic separation. As a result, most of the heavy contaminants—such as HDPE cups and lids, steel wire, glass, pebbles, or pieces of wood—were separated. Purified polyAl was then shredded to fractions below 8 × 8 mm and mixed for 20 min with a solvent at 120 °C. The dissolution of LDPE in a solvent was carried out in an Ichemad PSMH 22 F Ex1 horizontal paddle mixer, with a capacity of 1.6 m^3^ per batch. The solvent used for dissolution was a mixture of medium–low-boiling-point aliphatic esters. During mixing, the LDPE was fully dissolved. The resulting viscous suspension of aluminium particles in the polymer solution was then separated using a Flottweg Decanter centrifuge, model Z23 g. Decantation was performed at 120 °C with a g-force of 2000 g. In this process, the first product was separated—aluminium foil containing 55% residual solvent. The aluminium foil was then vacuum-dried from residual solvent in an AVA conical dryer, model HVW-VT 180, at 70 °C and 40 mbar(a). The LDPE solution, no longer containing aluminium foil or other contaminants, was transferred to a horizontal mixer of the same type as the one used for dissolution, but with a cooling jacket. During cooling, polyethylene powder precipitated from the solution, with the final temperature of the mixture being 70 °C. Upon precipitation, the powder was separated from the solvent by centrifugation and vacuum-drying using the equipment and parameters mentioned above. A sample of this powder (PE-T) was used as the carbon source for PHA production (Figure 2).

### 2.2. Media and Chemicals

Tryptone soya agar (TSA) and tryptone soya broth (TSB) were purchased from Lab M Ltd. (Lancashire, UK) and prepared according to the manufacturer’s protocol. Both TSB and TSA media contain peptone (20 g/L) and glucose (2.5 g/L). All basal salts used in the preparation of BSM (low nitrogen content) were obtained from BDH Chemicals Ltd. (Dorset, UK) and prepared accordingly: 1 L of distilled water, 1 g/L K_2_HPO_4_, 1 g/L KH_2_PO_4_, 1 g/L KNO_3_, 1 g/L (NH_4_)_2_SO_4_, 0.1 g/L MgSO_4_.7H_2_O, 0.1 g/L NaCl, and 10 mL/L trace elements. The trace elements solution contained 2 mg/L CaCl_2_, 2 mg/L CuSO_4_.5H_2_O, 2 mg/L MnSO_4_.5H_2_O, 2 mg/L ZnSO_4_.5H_2_O, 2 mg/L FeSO_4_, and 2 mg/L (NH_4_)_6_Mo_7_O_24_.4H_2_O. Ringer’s solution (Lab M, Lancashire, UK) was used as a saline solution for the analysis of viable cells during the cultivation process. To prepare this solution, a ¼-strength tablet was left to completely dissolve in 500 mL of deionised water with constant stirring. All media used in this study were sterilised by autoclaving (Priorclave Ltd., London, UK) for 15 min at 121 °C.

### 2.3. Microorganism

The bacterial strain *Cupriavidus necator* H16 (NCIMB 10442, ATCC 17699) was used for this investigation. The stock culture was freeze-dried and stored at −20 °C in the University of Wolverhampton’s culture collection. Before use, the culture was revived by inoculation into sterile TSB and kept in a shaker incubator (150 rpm) at 30 °C and for 24 h. In preparation for the synthesis of PHA, an inoculum from the revived broth culture was streaked onto TSA plates and incubated overnight at 30 °C. The prepared plates were used in further experiments.

### 2.4. Pre-Culture Conditions

Prior to experimental use, overnight broth cultures (starter) were aseptically prepared using the stock plates. Starter cultures were first prepared in four different 50 mL conical flasks, each containing 20 mL of TSB medium. Single colonies of *C. necator* from the initial streaked plate were aseptically inoculated into these flasks. All flasks were cultured aerobically at 30 °C in TSB medium for 24 h, with rotary shaking at 150 rpm (New Brunswick Scientific Co. Series 25 Incubator Shaker, Enfield, CT, USA). Furthermore, at the end of this incubation period, small samples of the microbial cultures were aseptically collected to perform a Gram staining analysis to ascertain that these cultures were free from any form of contamination.

### 2.5. Shake-Flask Production of PHA

In this study, 230 mL of each growth medium (TSB or BSM) was prepared in a 500 mL conical flask as a batch system. Next, 50 mL of TSB or BSM was transferred aseptically into 100 mL sterile beakers, and 0.50 g of PE-T was added. To ensure an even dispersion of samples in the TSB/BSM media, these beakers and their contents were sonicated (Bandelin Electronic Sonicator, Berlin, Germany) for 10 min at 0.5 active and passive intervals using a power of 70%. The sterility of the emulsified broths was checked by spread plating on TSA medium. Next, 50 mL of sterile, emulsified medium was aseptically transferred back into a 500 mL conical flask containing the reaming 180 mL of TSB or BSM medium, followed by the addition of a 20 mL starter culture to each flask to create high-cell-density cultures. Thus, the final volume of the batch culture was 250 mL, and the pH of each medium was adjusted to pH 7.0. Each shake-flask experiment was conducted in triplicate. All bacterial cultures were incubated for 48 h or 72 h in a rotary incubator (New Brunswick Scientific Co. Series 25 Incubator Shaker, Enfield, CT, USA), with constant rotation of 150 rpm and at a constant temperature of 30 °C. An experimental control was also set up under the same conditions in either TSB or BSM medium only, without the addition of the PE-T sample.

To monitor the growth patterns during the cultivation process, viable cell counts were conducted using the method described by Miles and Misra [25]. Following this method, 0.5 mL of sample was aseptically withdrawn from each growing culture at times 0, 3, 6, 9, 15, 21, 24, 27, 30, 40, 44, 48, 54, 65, and 72. Each sample was serially diluted from 10^−1^ to 10^−8^. In the next step, 20 µL of each dilution was aseptically pipetted onto TSA plates in triplicate. All plates were incubated for 24 h at 30 °C. The colonies observed afterwards were counted, and the obtained counts were expressed in Log_10_ CFU/mL^−1^.

### 2.6. Determination of Cell Dry Weight (CDW)

At the end of the 48 h incubation period, all flasks were removed from the shaker incubator. The culture broth was collected in 400 mL centrifuge tubes and centrifuged for 10 min at 4500 rpm using a Sigma 6-16 KS centrifuge. The obtained cell pellet was left overnight to freeze in a −20 °C freezer. Each frozen cell pellet was lyophilised for 48 h at −40 °C and 5 mbar using an Edward freeze-drier (Modulyo, Crawley, UK). The cell dry weight (CDW) was determined by weighing out the obtained dried biomass.

### 2.7. PHA Extraction and Purification Procedure

To extract the PHA, the dried biomass obtained from each flask was placed in an extraction thimble, covered with cotton wool, transferred to a Soxhlet extraction set up with HPLC-grade chloroform (Sigma-Aldrich, St. Louis, MO, USA), and left to run for 48 h. As part of a chloroform–biopolymer mixture, the PHA extracted was collected. Using a rotary evaporator set to 50 °C, leftover chloroform present in the mixture was evaporated, leaving behind the extracted PHA gathered in a 250 mL round-bottomed flask. In order to further purify the obtained biopolymer, PHA was precipitated in n-hexane. The precipitation was conducted in the round-bottomed flask. The flask was swirled gently to dislodge any PHA stuck to the flask, and the dispersion formed afterwards was filtered with a filter paper (Whatman No. 1 paper, Whatman Laboratory, Cardiff UK) to separate the polymer from the hexane solution. The purified polymer was left in a fume cupboard to dry, and later formed a thin PHA film. The PHA yield was determined afterwards using Equation (1):(1)$$\mathrm{PHA}\;\mathrm{yield}\;\left(\%\right)=\left(\frac{\mathrm{Weight}\;\mathrm{of}\;\mathrm{extracted}\;\mathrm{polymer}\;\left(\mathrm{WPHA}\right)}{\mathrm{Cell}\;\mathrm{dry}\;\mathrm{weight}\;\left(\mathrm{CDW}\right)}\right)\times 100$$

### 2.8. PE Utilisation

The supernatant resulting from centrifuging the culture broth was recovered and filtered using a sieve to obtain the residual PE-T sample. The residual carbon source was left to concentrate to a constant mass. The carbon source utilisation was calculated using Equation (2):(2)$$\mathrm{Carbon}\;\mathrm{source}\;\mathrm{utilization}\;\left(\%w/w\;\right)=\frac{\mathrm{Initial}\;\mathrm{PE}-T\;–\;\mathrm{Residual}\;\mathrm{PE}-T}{\mathrm{Initial}\;\mathrm{PE}-T}\times 100$$

### 2.9. Data Analysis

Results recorded during the experiments—such as viable counts and PHA yields—were statistically analysed by analysis of variance (ANOVA) using Microsoft Excel 2016 statistical software, and the differences between means were compared using the least significant differences (LSD) at 5% level of probability (*p* ≤ 0.05).

### 2.10. PE-T and PHA Characterisation

#### 2.10.1. Fourier-Transform Infrared (FTIR) Spectroscopy

The PE-T and PHA polymers obtained after extraction were characterised using a Bruker FTIR spectrometer (UK) with an alpha platinum single-reflection diamond attenuated total reflectance (ATR) module. The sample loading jet was cleaned with acetone, after which a thin layer of the polymer sample was loaded and secured with the loading knob. The absorbance spectra of the samples were recorded at wavenumber values between 4000 and 500 cm^−1^, and 10 scans were carried out.

#### 2.10.2. Gel Permeation Chromatography (GPC) Analysis of PE-T

The GPC experiments for plain LDPE—used for Tetra Pak^®^ production—and PE-T were conducted in a 1,2,4-trichlorobenzene solution (with antioxidant) at 160 °C, and at a flow rate of 1 mL/min, using an Agilent PL GPC220 Integrated HT GPC system with Agilent PLgel Olexis guard plus 3 × Olexis, 30 cm, 13 μm columns in series, and a refractive index detector (with differential pressure and light scattering). A single solution of each sample was prepared by adding 15 mL of eluent to 15 mg of sample and heating at 190 °C for 20 min while shaking. The solutions were allowed to cool to 160 °C, and were then filtered through a 1.0 μm glass-fibre mesh. The solutions were filtered directly into autosampler vials, and injection of samples was carried out automatically. The GPC system used was calibrated using a series of Agilent/Polymer Laboratories EasiVial PS-H polystyrene calibrants, but a mathematical procedure involving the use of viscosity constants from the literature was applied to the calibration to express the results in the same manner as for linear polyethylene.

#### 2.10.3. GPC Analysis of PHA

The number-average molar mass (M_n_) and the molar mass distribution index (M_w/_M_n_) of the obtained PHA samples were determined by GPC experiments conducted in a chloroform solution at 35 °C and at a flow rate of 1 mL/min using a Viscotek VE 1122 (Malvern, Worcestershire, UK) pump with two Mixed C PLgel Styragel columns (Agilent, Santa Clara, CA, USA) in series, and a Shodex SE 61 RI detector (Showa Denko, Munich, Germany). A volume of 10 μL of a chloroform sample solution (concentration 0.5% m/V) was injected into the system. The instrument was calibrated using polystyrene standards with low dispersity.

#### 2.10.4. Nuclear Magnetic Resonance (NMR)

NMR analysis was carried out using a 600 MHz Bruker Avance II (Bruker, Rheinstetten, Germany) and a 400 MHz JEOL NMR spectrometer JNM-ECZ400R/M1 (Akishima, Tokyo, Japan). Deuterated chloroform was used as the solvent, and tetramethylsilane (TMS) as the internal standard.

#### 2.10.5. Electrospray Tandem Mass Spectrometry Analysis (ESI-MS/MS)

Electrospray tandem mass spectrometry analysis was performed using a Finnigan LCQ ion trap mass spectrometer (Thermo Finnigan LCQ Fleet, San Jose, CA, USA). PHA obtained from each medium’s conditions was partially degraded to lower-mass PHA oligomers, which were then dissolved in a chloroform/methanol system (1:1 *v/v*) [22]. The solutions were introduced into the ESI source by continuous infusion using the instrument syringe pump at a rate of 5 µL/min. The LCQ ESI source was operated at 4.5 kV, and the capillary heater was set to 200 °C; the nebulising gas applied was nitrogen. For ESI-MS/MS experiments, the ions of interest were isolated monoisotopically in the ion trap, and were activated by collisions. The helium damping gas that was present in the mass analyser acted as a collision gas. The analysis was performed in the positive-ion mode.

#### 2.10.6. ^14^C analysis (AMS Technique)

Combustion of organic samples was performed in closed (sealed under vacuum) quartz tubes with CuO and Ag wool at 900 °C for 10 h. The obtained gas (CO_2_ + water vapour) was then dried in a vacuum line and reduced with hydrogen (H_2_) using 2 mg of iron powder as a catalyst. The ^14^C content in the carbon samples was determined using a “Compact Carbon AMS” spectrometer [26], by comparing the intensities of ionic beams of ^14^C, ^13^C, and ^12^C measured for each sample and for standard samples (the modern standard “Oxalic Acid II” and the standard of 14C-free carbon “background”). The ^14^C content in each sample was then expressed as a percentage of modern carbon (pMC), calculated using correction for isotopic fractionation [27].

## 3. Results

### 3.1. Characterisation of Carbon Source (PE-T)

The molecular structure of the units of PE from Tetra Pak^®^ (PE-T) was studied using FTIR spectroscopy. The characteristic absorbance bands for PE (2910 cm^−1^, 2845 cm^−1^, 1470 cm^−1^, and 718 cm^−1^) were observed in the FTIR spectrum shown in Figure 3, and were consistent with observations made in [28]. The observed frequencies of the PE-T material and its possible assignments are also represented in Table 1.

A split in the 720 cm^−1^ peak, possibly due to the crystallinity of the PE-T material, was also observed. The spectrum also shows a C=O group band at 1708 cm^−1^, indicating the possible presence of carboxylic acid, peroxides, esters, or ketones around this region, which could contribute to improving the hydrophilicity of the PE-T material. Consequently, the different groups observed in this spectrum show that the PE-T sample should be a suitable and accessible carbon source for PHA production.

The GPC chromatograms (Figure 4) show a high molecular weight for both PE-T and plain LDPE, although the molar mass of plain LDPE was slightly higher than that of PE-T. The average molar masses and dispersity of these materials are shown in Table 2. The observed lower dispersity of PE-T (in comparison to LDPE) may be caused by some fractionation during the extraction process.

### 3.2. Microbial Growth Analysis

The effects of two different liquid growth media—nitrogen-rich tryptone soya broth (TSB) and nitrogen-limited basal salt medium (BSM), both supplemented with PE-T—were investigated. The growth of *C. necator* in TSB or BSM supplemented with PE-T was monitored during the 48 to 72 h incubation period by observing the viable counts (Log_10_CFU/mL) at times of 0, 3, 6, 9, 15, 21, 24, 27, 30, 40, 44, 48, 54, 65, and 72. All performed bacterial experiments were started with high cell numbers to create high-cell-density cultures, which should promote the production of non-growth-linked metabolites such as PHAs. All growth patterns of *C. necator* obtained from this analysis are shown in Figure 5. The mean value of each viable cell count was calculated. Replicates were averaged and statistically analysed.

Figure 5 shows microbial cultivation with and without (controls) PE-T added to both TSB and BSM after 72 h of incubation. An increase in CFU/mL in the first 24 h of incubation was observed in all fermentation conditions, with the highest cell increase observed in TSB with PE-T. Subsequently, after 24 h, a gradual decline in cell viability was observed for all growth conditions. Under the same growth conditions, quite a similar growth pattern to that seen in this study was previously reported by Ekere et al. [23] when low-density PE fragments were used as the sole carbon source, although in their study better cell growth was observed with BSM. The results presented in this study (Figure 5) show a similar pattern for both controls and both media supplemented with PE-T. Therefore, it can be concluded that use of PE-T initiated an increase in cell growth, and had no inhibitory effect on the viability of C. necator irrespective of the cultivation conditions. When the cultivation time was stretched to 72 h (Figure 5), a continuous decline in cell viability was observed for all fermentation conditions. This indicates that the culture conditions were no longer sustainable to facilitate cell viability, and that 48 h of batch culture would be an ideal optimised period to initiate cell growth and, possibly, PHA accumulation.

The obtained biomass from all of the experiments described above with PE-T as a sole or additional carbon source, as well as controls (without PE-T), was tested for accumulation of PHA. The data in Table 3 show that the use of PE-T significantly affected PHA production (*p* ≤ 0.05). Cells cultured in TSB with the addition of PE-T for 48 h (Figure 5) showed the highest amount of stored polymer (40% PHA of CDW), and no stored PHA was observed in the 48 h culture of BSM with PE-T. Significantly lower PHA production was seen in control cultures cultivated in TSB alone (1.5% PHA of CDW), regardless of carbon sources present in TSB in the form of peptone and glucose. After a 72 h culture period (Figure 5), no stored polymer was obtained from cells cultured in either TSB or BSM with PE-T. This result, as presented in Table 3, also indicates that the accumulation of PHA in TSB alone is not simultaneous to the growth of *C. necator* in TSB. This further indicates the promising effect PE-T has on the accumulation of PHA by *C. necator* when cultivated in nitrogen-rich media.

The amount of carbon source utilised by *C. necator* was determined by measuring the residual PE-T content after the 48 h of cultivation using an air process (Table 4). Higher carbon utilisation was observed in TSB with PE-T and in BSM medium with PE-T (Table 4). The results show that *C. necator* utilised the PE-T in both growth media, but was able to accumulate PHA only in the TSB medium (Table 3). The addition of PE-T to BSM medium as the sole carbon source was thus used to maintain viability, but not for PHA accumulation.

### 3.3. PHA Identification and Characterisation

The PHA extracted from TSB with PE-T was characterised using FTIR, NMR, GPC, ESI-MS, and ASM carbon dating. The results of FTIR analysis of the polymer synthesised by *C. necator* with PE-T in TSB are shown in Figure 6. The FTIR spectra confirmed the indicative peak of a typical polyester group detected at 1722 cm^−1^, corresponding to the (C=O)-stretching band of the ester carboxylic acid group. The strong absorption band at 1722 cm^−1^ indicates that most of the produced polymers have carboxylic acids, as opposed to the possible weak presence of carboxylic acids observed in the PE-T starting material. Other relevant bands observed at 3389 cm^−1^, 2977 cm^−1^, 2926 cm^−1^, and 2848 cm^−1^ corresponded to hydroxyl (O-H), methylene (CH_2_), methyl (CH_3_), and CH group stretching, respectively. The strong bands in the range of 1454–1000 cm^−1^ (assigned to C-O-C stretching and C-C, C-O, CH_2_, and CH_3_ bending) were observed in the PHA sample, but were absent in the PE-T starting material. There is the likelihood that these bonds suggest the PHBV nature of this PHA sample. Another report [23] also described similar observations for the production of P(HB-co-HV). The differences between the FTIR spectra of the starting PE-T material and the FTIR spectra of the resulting PHA sample are highly noticeable, and depict the purity of the PHA sample produced from PE-T.

The chemical structure of the PHA from PE-T was further analysed by ^1^H-NMR spectrometry. The results of this analysis can be seen in Figure 7. The spectrum revealed resonance signals from protons associated with 3-hydroxybutyrate (3HB) at 1.28 ppm (CH_3_), 5.32 ppm (CH), and 2.64 ppm (CH_2_). The presence of 3-hydroxyvalerate (3HV) repeating units was also detected; the signals at 0.90 ppm and at 5.19 ppm correspond to the protons of a methyl group (CH3) of the side chain and methine group of the main chain, respectively. The signals of the protons of CH_2_ groups of HV and HB units were overlapped [29]. The molar composition of the obtained PHBV copolymer was determined from the 1H NMR spectrum, based on the integration of the signals corresponding to the protons of the CH3 groups of HB and HV units 1 and 4, respectively (Figure 7). The calculated content of HV units in the synthesised PHBV copolymers was not more than 8 mol%.

Further characterisation of the PHA obtained was carried out to determine the molecular weight distributions of the polymer using the GPC technique. As revealed in the GPC chromatogram, the weight-average molar mass (Mw), number-average molar mass (Mn), and dispersity index (DI) of the purified PHA were 458,000 g/mol, 196,000 g/mol, and 2.3, respectively. From these results, it is certain that using PE-T as an additional carbon source could generate PHAs with high molecular masses.

The ESI-MS technique was used for an advanced and detailed structural characterisation of the PHA oligomer chains obtained from partial degradation of the PHA polymer [22]. The ESI-MS spectrum of the selected PHA oligomer, acquired in positive-ion mode, is presented in Figure 8. The spectrum shows singly charged ions that are grouped in numerous clusters due to their different degrees of oligomerisation and composition. The signals belonging to the main series occurring in the spectrum, with a mass difference of 86 Da between the *m/z* values (i.e., the molecular mass of 3-hydroxybutyrate (3-HB) units), correspond to the potassium adducts of PHB oligomers terminated by crotonate and carboxylic end groups. The series of ions moved by 14 to the main series were also observed. For further verification of the individual structure of the oligomer chains belonging to the clusters, tandem mass spectroscopy (ESI-MS/MS) was performed for the selected ions isolated from the ESI-MS spectrum in Figure 8.

This ESI-MS/MS spectrum displayed in Figure 9 was attained from the precursor ions at *m/z* 1027. The fragmentation of the ion at *m/z* 1027 led to the formation of three series of product ions. As previously indicated in the spectrum displayed in Figure 8, the first main series of ions seen in Figure 9 (*m/z* 941, 855, 769, 683) also corresponds to the potassium adducts of PHB oligomers [22]. The second set of product ions at *m/z* 927 and 841 occurred due to the expulsion of 2-pentenoic acid (100 Da), confirming the presence of 3-hydroxyvalerate units in addition to the detected 3-HB units. For the formation of the third, smallest series of product ions at *m/z* 827, 741, 655, 569, and 483, a loss of 2-hexenoic acid (114 Da) was detected and assigned to HH units along the PHA oligomer chain. Thus, based on the results of fragmentation, it can be assumed that the isolated ions at *m/z* 1027 represent potassium adducts of the PHA oligomer terminated with carboxyl and unsaturated end groups, and contain along the chain three randomly distributed HV units or one HV unit and one HH unit with the structure ([HB_8_HV_3_ + K]^+^) or ([HB_9_HV HH + K]^+^), respectively. However, the presence of HH units was not confirmed by NMR analysis.

The results obtained from ^14^C analysis of PHA isolated from *C. necator* with PE-T are shown in Table 5.

## 4. Discussion

The use of post-consumer waste products as feedstock for PHA production could enhance its process economics and market implementation. In addition, employing the PE component in waste Tetra Pak^®^ as feedstock for PHA production would be an economical and excellent Tetra Pak^®^ recycling technique for curbing Tetra Pak^®^ waste streams while producing value-added PHA biopolymer. One major attribute of *C. necator* is its ability to accumulate PHA from various substrates, including glucose and fatty acids [30]. Here, the metabolism of this strain was evaluated in terms of its ability to produce PHA by utilising PE-T from Tetra Pak^®^ waste as the sole or an additional carbon source.

Generally, LDPE is an amorphous polymer with short branches comprising one or more co-monomers (such as 1-octene and 1-hexene) [31]. During sonication, the PE molecules are prevented from stacking closely together, due to its branching system, thus making the tertiary carbon atoms present at the branch sites more accessible and susceptible to microbial attack [31]. As also revealed by the FTIR analysis, and previously reported by Ojeda et al. [32], some carbonyl and hydroperoxide groups may also be present in the PE polymer, and could further improve the accessibility of the PE-T substrate. These short side chains may have been consumed first by *C. necator* to maintain viability. In addition, the growth curve (Figure 5) shows that PE-T was consumed during incubation, as indicated by the increase in biomass when TSB was supplemented with PE-T. Furthermore, the ability of *C. necator* to remain viable in BSM limited in nitrogen content also suggests that PE-T substrates exhibit no antimicrobial properties and, thus, have no negative effects on the growth of *C. necator* in either TSB or BSM. This finding is consistent with a previous report by Gregorova et al. [33]. The cultivation of *C. necator* in all performed experiments was started with high cell numbers to create high-cell-density cultures. This should quickly create a stress condition, resulting in slow cell growth or even growth inhibition whilst enhancing the production of non-growth-linked metabolites such as PHAs in both non-limited (TSB) and limited (BSM) nitrogen conditions. It is known that PHAs are produced by bacteria under stress conditions [34,35]. In high-cell-density cultivation, cells are subjected to oxygen limitation (as cells compete for oxygen) and, furthermore, to the accumulation of toxic metabolites during incubation. Therefore, as a result, *C. necator* showed a small increase in the cell numbers in TSB and/or BSM with Tetra Pak^®^ and without it (controls).

It can be assumed from this study that PE-T could be used as a carbon source by *C. necator.* While limited growth was observed with both TSB and BSM media, PHA accumulation occurred in TSB medium only (Table 3). As observed in previous studies [34,35], the limited nitrogen content in BSM hinders its influence on PHA accumulation. The maximum PHA yield (40% CDW) obtained in this study was achieved after 48 h of growth in TSB supplemented with PE-T. The purification process of PE conducted in this study obviously facilitated the easier utilisation of PE-T by *C. necator*, resulting in higher PHA accumulation. The percentage PHA yield per cell dry weight obtained for PE-T (40% CDW) was considerably higher than in PE studies recently published by Ekere et al. [23] (29% CDW), Radecka et al. [34] (33% CDW), Guzik et al. [18] (25% CDW), and Johnston et al. [30] (32% CDW), when LDPE, oxidised PE wax, PE pyrolysed wax, and non-oxygenated PE wax, respectively, were used as carbon sources for aerobic PHA production. The compositional changes that may have occurred during the PE-T purification process could have facilitated microbial attachment and, thus, be responsible for the increased PHA yield in our study. Moreover, no form of contamination was observed in the PHA produced.

It is important to evaluate the biological contents of microbial bioplastics sourced from petroleum-based carbon resources to promote the use of bioplastics, reduce CO_2_ emissions into the atmosphere, and distinguish carbon atoms obtained from petroleum and those obtained from biomass [36,37]. Carbon-14 ratio measurement by AMS is the primary evaluation method commonly used for this purpose. The accuracy of this evaluation method has been previously demonstrated [36,37]; hence, its application in this study. The ^14^C method provided insight into the bio-derived fractions of the PHA obtained from this study. Employing radiocarbon analysis for this purpose was based on the large difference in ^14^C isotopic signatures between the petroleum-derived component (^14^C free) and the biogenic component of carbon-based materials [36]. This relies on the mechanism that modern biomass carbons comprise a certain (although very low) level of ^14^C, whereas petroleum carbons do not contain ^14^C [38]. Thus, by measuring the carbon-14 content of a bioplastic product derived from petrochemical sources, a ratio between carbon originating from the bio-based component (i.e., modern carbon) and that originating from the petroleum-derived components (i.e., old carbon) can be derived [39,40]. The AMS results showed that the starting purified LDPE sample (PE-T) had an old carbon content of 99.5%, and in the case of PHA obtained from TSB supplemented with PE-T, almost 97% of its carbon content was modern carbon derived from direct synthesis by *C. necator*, whereas only 3% was derived from old carbon present in the PE component of the Tetra Pak^®^ material. This further proves the potential of the bioconversion process proposed in this study for waste Tetra Pak^®^ packaging materials, and its use for PHA production.

More recent studies of microbial utilisation of PE polymers showed the use of PE polymer that had been subjected to some level of pretreatment technology [18,30,34]. In our study, however, no pretreatment technique was employed. Irrespective of the absence of a PE pretreatment technique, the results obtained in our study suggest that the purification process leading to the formation of LDPE powder was enough to make the carbon monomers accessible for improved microbial utilisation. This further makes the PE-T sample recovered from Tetra Pak^®^ a suitable and economical carbon source for PHA production.

The complexity of the PE-T sample can affect the structural composition of the monomer and, consequently, the properties of the accumulated PHA polymer. To determine the chemical structure of the monomeric units, as well as the composition of co-monomers in the PHA polymer synthesised by *C. necator* in this study, electrospray mass spectrometry (ESI-MS) was used. With the aid of this technique, it was confirmed that the PHA copolymer obtained from bacterial cultures with PE-T from Tetra Pak^®^ comprised 3-hydroxybutyrate, 3-hydroxyvalerate, and 3-hydroxyhexanoate co-monomeric units. Thus, the structure of PHA synthesised in this study with the aid of *C. necator* was similar to that of biopolyesters previously prepared by some of our team using other PE-based carbon sources [23,34].

## 5. Conclusions

In this study, we extensively demonstrated a novel Tetra Pak^®^ recycling method via a bioconversion process that uses PE-T recovered from Tetra Pak^®^ waste in the batch production of high-value biodegradable PHA. According to the principles of creating a circular economy, this study demonstrated that the mechanical grinding and separation process using the “solvents method” could be exploited for recovery of PE—and potentially other polymers—from waste materials. The microbial synthesis of high-molecular-weight PHA with 3-HB, 3-HV, and 3-HH co-monomeric units from TSB medium supplemented with recovered PE-T makes the polymer produced of special interest in various medical and industrial applications. Thus, this study offers a promising path for designing a sustainable biotechnology process, where biodegradable polymers can be produced using the post-consumer waste stream as a source of raw materials.

## 6. Patents

Partial degradation of PHA is the subject of EU patent entitled “Process for Controlled Degradation Of Polyhydroxyalkanoates And Products Obtainable There From.” Ep 2 346 922 B1. International publication number: WO 2010/044112 (22.04.2010 Gazette2010/16). Date of filing: 15.10.2008. Date of publication of application: 27.07.2011.

## Figures and Tables

**Figure 1 polymers-14-02840-f001:**
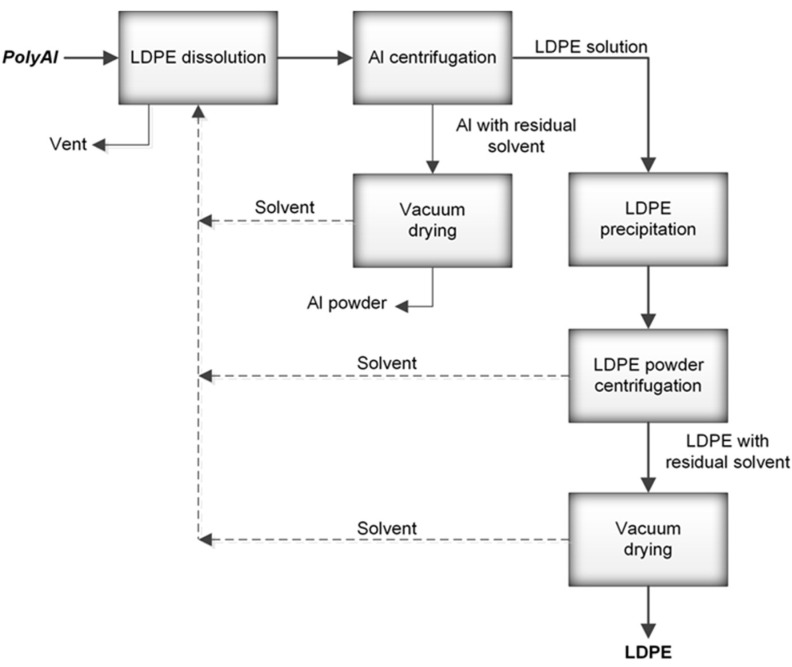
Schematic diagram of the processing method used for obtaining polyAl from waste Tetra Pak^®^.

**Figure 2 polymers-14-02840-f002:**
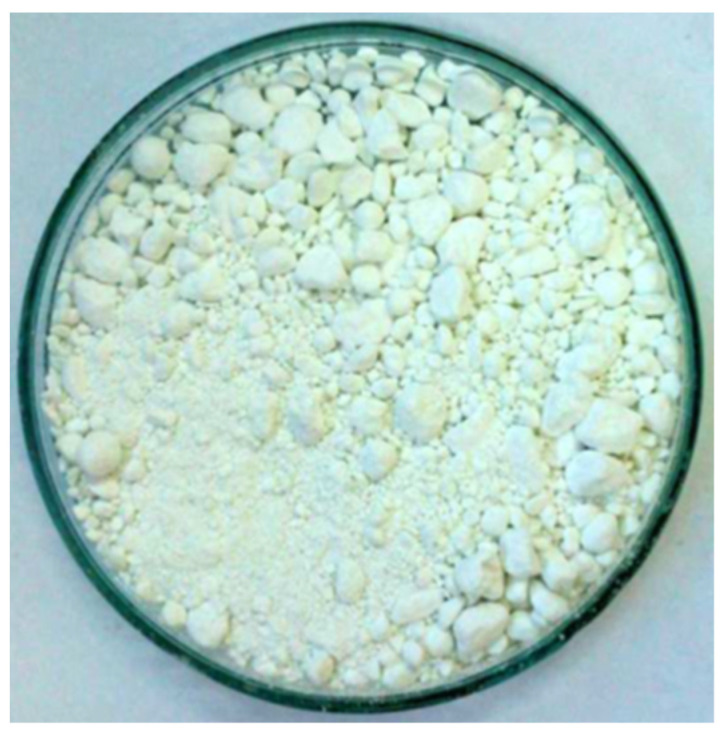
LDPE powder (PE-T) recovered from Tetra Pak^®^ packaging after precipitation, separation, and drying.

**Figure 3 polymers-14-02840-f003:**
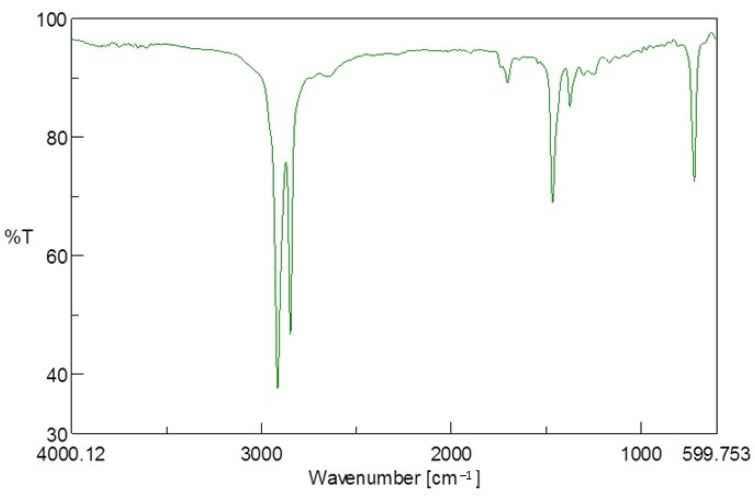
FTIR spectrum of PE-T used as a sole (BSM) or additional (TSB) carbon source in this study.

**Figure 4 polymers-14-02840-f004:**
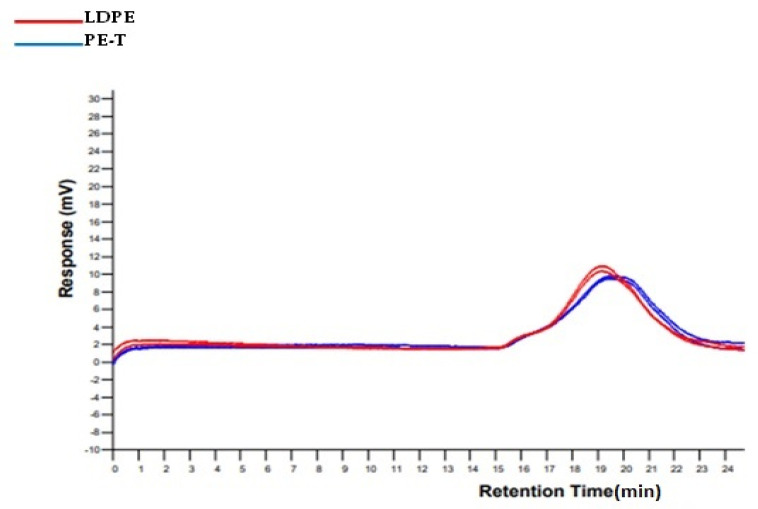
The GPC refractive index chromatograms for duplicate runs of PE-T and plain LDPE.

**Figure 5 polymers-14-02840-f005:**
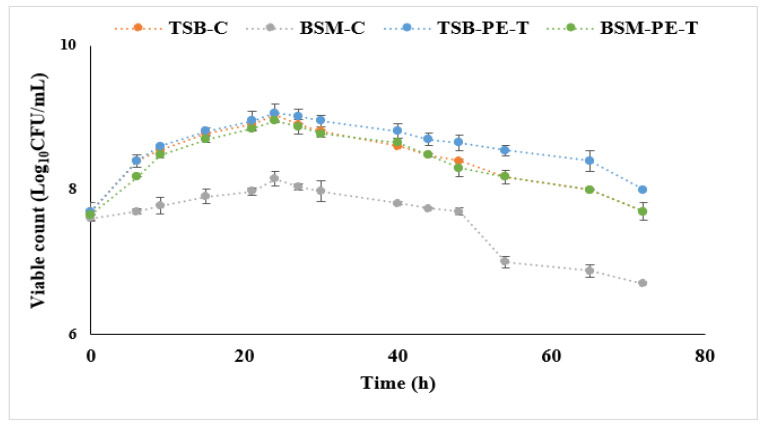
Growth observation of *C. necator* H16 with 0.5 g of PE-T in either TSB or BSM medium after 72 h of incubation at 30 °C. Viable count (Log_10_ CFU/mL) data points at times of 0, 3, 6, 9, 15, 21, 24, 27, 30, 40, 44, 48, 54, 65, and 72 h are the mean values of triplicate experiments (n = 3). Error bars represent the standard error of the mean values.

**Figure 6 polymers-14-02840-f006:**
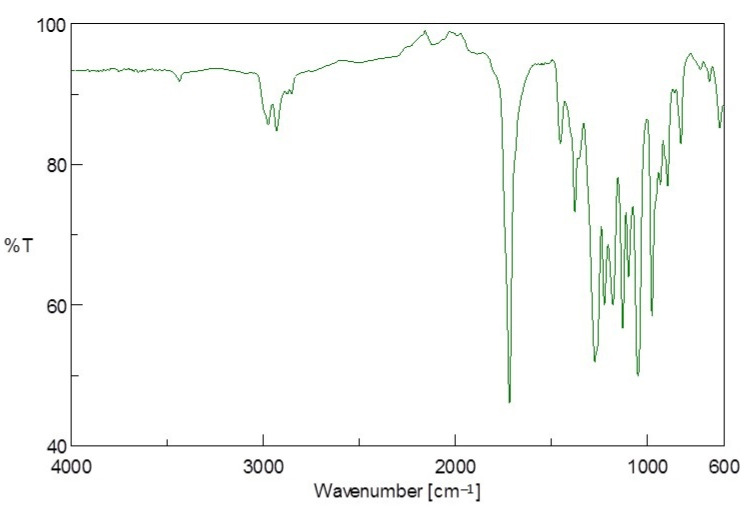
FTIR spectrum of PHA obtained from TSB supplemented with PE-T.

**Figure 7 polymers-14-02840-f007:**
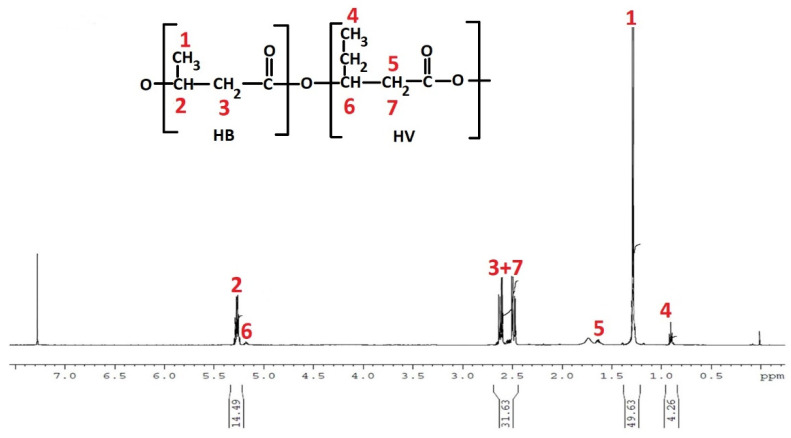
^1^H-NMR of PHA produced by *C. necator* utilising PE-T as an additional carbon source in TSB medium.

**Figure 8 polymers-14-02840-f008:**
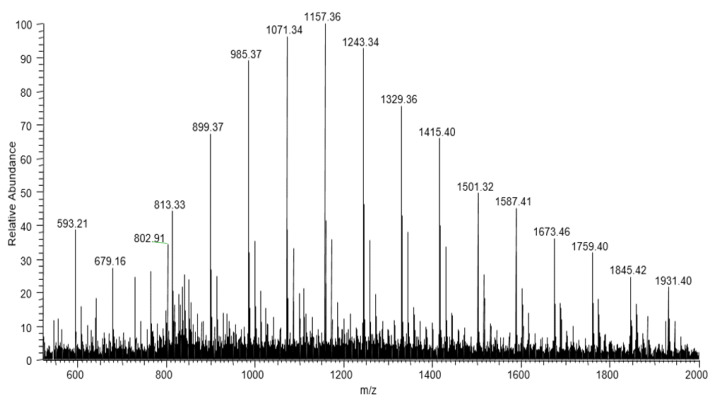
The ESI-MS (positive-ion mode) spectrum of PHA oligomers, obtained via partial thermal degradation of the biopolyester produced by *C. necator* H16 in TSB using PE-T as an additional carbon source.

**Figure 9 polymers-14-02840-f009:**
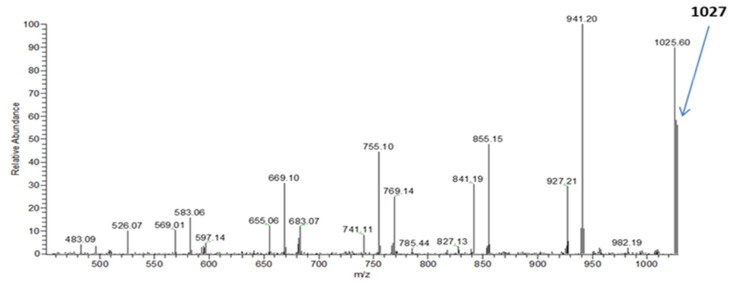
The ESI-MS/MS spectrum (positive-ion mode) of the PHA oligomers’ parent ion at *m/z* 1027.

**Table 1 polymers-14-02840-t001:** The major absorptions of PE-T observed in the IR region, and their assignments.

Band (cm^−1^)	Assignment	Intensity
2910	CH_2_ asymmetric stretching	Strong
2845	CH_2_ asymmetric stretching	Strong
1708	C=O stretching	Weak
1470	Bending deformation	Strong
1375	CH_3_ symmetric deformation	Weak
1099-599	C-C stretching vibrations	Weak
718	C-C stretching vibrations	Strong

**Table 2 polymers-14-02840-t002:** The average molar masses and dispersity of both PE-T and plain LDPE.

Sample	Mw (g/mol)	Mn (g/mol)	Mw/Mn
PE-T	74,000	13,300	5.6
Plain LDPE	83,000	13,100	6.4

**Table 3 polymers-14-02840-t003:** PHA synthesis by *C. necator* in TSB/BSM alone and TSB/BSM supplemented with PE-T after 48 h and 72 h of incubation at 30 °C. Percentage yields of PHA are the mean values of three replicates. ND: not detected.

48 h Growth Average Medium CDW (g/L)	PHA Yield (%/*w/w*)	72 h Average CDW (g/L)	PHA Yield (% *w/w*)
TSB-C 0.865	1.50 ± 0.03	0.490	ND
BSM-C 0.073	ND	0.040	ND
TSB-PE-T 0.820	40.0 ± 0.44	0.510	ND
BSM-PE-T 0.087	ND	0.060	ND

**Table 4 polymers-14-02840-t004:** PE-T utilisation by *C. necator* grown in TSB or BSM medium after 48 h of incubation.

Growth Medium	Initial PE-T (g)	Average Residual PE-T (g)	Carbon Source Utilisation (*w/w*%)
TSB with PE-T	0.50	0.32	36 ± 0.08
BSM with PE-T	0.50	0.30	40 ± 0.20

**Table 5 polymers-14-02840-t005:** ^14^C results for PHA samples obtained from cultures with and without PE-T, compared to purified PE-T only.

Sample	Modern Carbon (%)	Old Carbon (%)
Purified PE-T	0.47 ± 0.11	99.5 ± 0.11
Control PHA	100 ± 0.3	<0.3
PHA from TSB-PE-T	96.73 ± 0.32	3.27 ± 0.3

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
