# Peer review of "Bioconversion Process of Polyethylene from Waste Tetra Pak® Packaging to Polyhydroxyalkanoates"

_polymers, 2022, doi:10.3390/polym14142840_

Round 1

Reviewer 1 Report

Dear all,

 Greetings

 The authors have performed a good  the results of a novel recycling method for waste Tetra Pak® pack- 22 aging materials, but this paper can be accepted for publication in Polymers, after fixing all these points (Minor Revisions)

Ref : polymers-1782950

Title: Bioconversion Process of Polyethylene From Waste Tetra Pak® 2 Packaging to Polyhydroxyalkanoates

1) Title ok

2) Abstract: please add the final percentage of your conversion system

3) Keywords: OK

4) Introduction :

  • terials [X]; paperboard [Y], aluminium foil[Z], and low-density polyethylene (LDPE)[W], please add reference and do it for all your properties !!!

5) Materials and Methods: ok 

6) Results: 

  • FTIR is enough or you should add other spectroscopic methods 
  • Please compare figures 3 and 6 results

7) Conclusion : ok but need more detail about the high conversion of your system, could we used for other polymers too?

8) References:

please update them add some of 2021 and 2022

 fellow the journal template 

With regards

Author Response

Reviewer 1

Dear all, Greetings. The authors have performed a good  the results of a novel recycling method for waste Tetra Pak® pack- 22 aging materials, but this paper can be accepted for publication in Polymers, after fixing all these points (Minor Revisions). Ref : polymers-1782950. Title: Bioconversion Process of Polyethylene From Waste Tetra Pak® 2 Packaging to Polyhydroxyalkanoates.

Dear reviewer, we very much appreciate your comments, amendments as suggested have been added to the manuscript (highlighted) and explained below:

1) Title ok

2) Abstract: please add the final percentage of your conversion system

Response: Final percentage for each parameter has been added (see lines 27-29 and 34-35)

3) Keywords: OK

4) Introduction :

  • Materials [X]; paperboard [Y], aluminium foil[Z], and low-density polyethylene (LDPE)[W], please add reference and do it for all your properties !!!

Response: References have been added. (See lines 50, 57,59, 61,62,84,87 and 110)

5) Materials and Methods: ok 

6) Results: 

  • FTIR is enough or you should add other spectroscopic methods 

Response: We believe that for this study FTIR was a good method to use. FT-IR was employed as a fingerprinting technique to assess  PE-T and PHA produced. However, for detailed analysis of produced PHAs we also used NMR and ESI-MS.

  • Please compare figures 3 and 6 results

Response: This has been compared in lines 424-427, 429-431 and 433-435)

7) Conclusion: ok but need more detail about the high conversion of your system, could we used for other polymers too?

Response: The high conversion of our system and its potential has been addressed in lines 594-599.  Potentially it could be used for other polymers, so we might check it in our future studies.

8) References: please update them add some of 2021 and 2022

Response: References has been updated (see highlighted references)

 fellow the journal template

Response: Journal template has been implemented in all references listed 

Reviewer 2 Report

The work entitled: "Bioconversion Process of Polyethylene From Waste Tetra Pak® Packaging to Polyhydroxyalkanoates" presents an interesting aspect of the use of a novel recycling method for waste Tetra Pak® packaging materials, along-side its potential for producing value-added PHA, and the ability of 14C analysis in validating this bioconversion process. it can be accepted for publication in the journal after the authors considered minor errors.

1) Abstract present a clear purpose and scope of the work, provide a short research methodology but should present the results obtained through the study well and present the most important conclusions. It needs to be corrected.

2) the Figures resolution should be improved.

3) The conclusion should be more concise, generalising and summarizing.

4) Please revise number of references in text with the list of references section.

5) Please follow the instructions for authors of the journal.  

Author Response

Reviewer 2

The work entitled: "Bioconversion Process of Polyethylene From Waste Tetra Pak® Packaging to Polyhydroxyalkanoates" presents an interesting aspect of the use of a novel recycling method for waste Tetra Pak® packaging materials, along-side its potential for producing value-added PHA, and the ability of 14C analysis in validating this bioconversion process. it can be accepted for publication in the journal after the authors considered minor errors.

Dear reviewer, we very much appreciate your comments, amendments as suggested have been added to the manuscript (highlighted) and explained below:

1) Abstract present a clear purpose and scope of the work, provide a short research methodology but should present the results obtained through the study well and present the most important conclusions. It needs to be corrected.

Response: Thank you for you comments.  Abstract was revised (see lines 27-29, 34-35)

2) the Figures resolution should be improved

Response: All figures are now saved in as a high-resolution files. This should improve the quality of the images.

3) The conclusion should be more concise, generalising and summarizing.

Response: Conclusion has been revised.

4) Please revise number of references in text with the list of references section.

Response: This has been revised accordingly.

5) Please follow the instructions for authors of the journal

Response: Authors format has been revised (see lines 4-6)

Reviewer 3 Report

The concept of upcycling should be introduced in the Introduction.

Further details about the separation of LDPE from aluminium in the PolyAl waste should be added. Please consider introducing the relevant information of the equipments (model, brand, etc.) and conditions used in each stage.

Please indicate details about the extraction process applied to separate the PHA from the bacterial walls

The mol% content of 3HB or 3HV in the PHBV (one can assume this was produced) could be determined from the NMR tests.

However, can authors fully state that only one type of PHA was generated?

Perhaps a DSC analysis can be useful to ascertain the type(s) of PHA produced by the obsevation of the melting point

Author Response

Reviewer 3

Dear reviewer, we very much appreciate your comments, amendments as suggested have been added to the manuscript (highlighted) and explained below:

The concept of upcycling should be introduced in the Introduction.

Response: Concept is introduced in lines 94-98 and 112-113

Further details about the separation of LDPE from aluminium in the PolyAl waste should be added. Please consider introducing the relevant information of the equipments (model, brand, etc.) and conditions used in each stage.

Response: We have expanded this part of the description with the suggested information.

however, due to company secrets, we cannot provide all information. (See lines 125-141)

Please indicate details about the extraction process applied to separate the PHA from the bacterial walls

Response: Details about Chloroform extraction process applied in this study has been explained in lines 226 - 242

The mol% content of 3HB or 3HV in the PHBV (one can assume this was produced) could be determined from the NMR tests.

Response: Thanks for this comment. Of course, the molar composition of the obtained PHBV copolymer can be easily determined from the 1H NMR spectrum, based on integration of the signals corresponding to the protons of CH3 groups of HB and HV units 1 and 4, respectively (Fig. 7). The calculated content of HV units in the synthesized PHBV copolymers was not more than 8 mol%. The manuscript has been updated with this information (See lines 447-451).

However, can authors fully state that only one type of PHA was generated?

Response: To determine the chemical structure of the PHA polymer synthesized by C. necator in this study the 1H NMR was applied. Additionally, to confirm the molecular-level structure of the obtained PHA copolymers multistage mass spectrometry ESi-MSn was used. The acquired ESI-MS spectrum of the PHA oligomers looks like a typical spectrum for random PHA copolymers. The spectrum shows singly charged ions which represent the individual copolymer chains grouped in numerous clusters due to their different composition and degrees of oligomerization. The signals within the individual clusters are moved by 14 Da due to the differences between molar masses of HV and HV units. Based on the results of fragmentation of the selected ions it was confirmed that those ions represent potassium adducts of the PHA oligomers terminated with carboxyl and unsaturated end groups and contains along the chain predominantly HB and randomly distributed HV and HH units, respectively. However, the low content of HH units was not confirmed by NMR analysis.

Perhaps a DSC analysis can be useful to ascertain the type(s) of PHA produced by the obsevation of the melting point

Response: To confirm the structure of the obtained PHAs, we used NMR and ESI_MS techniques, which confirmed that the obtained PHAs are random copolymers. Mass spectrometry, as a very sensitive technique, also confirmed the very low content of HH units. In the case of random copolymers, with a relatively small share of HV units, about 8 mol%, and a much lower share of HH units (not detectable by NMR), it is difficult to expect changes in the values of melting points.